# Inequalities in the distribution of National Institutes of Health research project grant funding

**Michael S Lauer[1]\*, Deepshikha Roychowdhury[2]**

[1]National Institutes of Health, Office of the Director, Bethesda, United States; [2]NIH Office of Extramural Research, Bethesda, United States

**Abstract** Previous reports have described worsening inequalities of National Institutes of Health (NIH) funding. We analyzed Research Project Grant data through the end of Fiscal Year 2020, confirming worsening inequalities beginning at the time of the NIH budget doubling (1998–2003), while finding that trends in recent years have reversed for both investigators and institutions, but only to a modest degree. We also find that career-stage trends have stabilized, with equivalent proportions of early-, mid-, and late-career investigators funded from 2017 to 2020. The fraction of women among funded PIs continues to increase, but they are still not at parity. Analyses of funding inequalities show that inequalities for investigators, and to a lesser degree for institutions, have consistently been greater within groups (i.e. within groups by career stage, gender, race, and degree) than between groups.

## Introduction

Over the past few years, there has been increasing interest (*Peifer, 2017*) in how the National Institutes of Health (NIH) funding support is distributed, with concern voiced by some that there may be excess concentration of support given to men and to the most well-funded late-career investigators. In a report (*National Institutes of Health, 2019*) issued by an NIH Working Group to the Advisory Committee to the Director (ACD), it was noted that "In biomedical science, power stems from who has access to awards. The Working Group heard repeatedly that the concentration of funding in a relatively small number of investigators (who are overwhelmingly white, cisgender, straight men) incentivizes universities to protect researchers bringing in high levels of grant funding".

Recently published literature has raised concerns regarding how NIH distributes funding support. One report (*Katz and Matter, 2020*) which focused on all 'R' grants found increasing inequality of funding support over 30 years (1985–2015). A research letter (*Oliveira et al., 2019*) found lower levels of support for grants in which women were identified as Principal Investigators. Other reports have documented disproportionate aging of the research workforce (*Blau and Weinberg, 2017*) and stresses particular to mid-career investigators (*Charette et al., 2016*); these reports are concerning given evidence that there is no correlation between research stage and scientific impact (*Sinatra et al., 2016*).

In 2017, the NIH considered imposing a cap (*Lauer, 2017a*) on individual-investigator research support through use of a 'Grant Support Index or GSI' (*Lauer, 2017b*) which classified grants according to mechanism (e.g. R01, P01, U54) rather than according to dollars. The GSI set a value of 7 for R01 grants, with lower values for 'smaller' mechanisms like R03 or R21 and greater values for mechanisms like P01 or U54. The proposed cap was set at 21, meaning that on average no investigator could be designated as PI on more than the equivalent of three R01 grants. The proposed cap was highly controversial (*Kaiser, 2017*) and was dropped in favor of a different approach (*Lauer et al., 2017*) that targeted funds directly toward early career investigators.

**\*For correspondence:**
Michael.Lauer@nih.gov

**Competing interests:** The authors declare that no competing interests exist.

Here, we present updated data on distribution of NIH support for principal investigators ('PIs', keeping in mind that NIH issues awards to institutions [*Lauer, 2018*], not to individual scientists) with particular attention to career stage, gender, race, and degree. We focus on research project grants ('RPGs') as these comprise close to 80% of all NIH extramural research funding; we can also assess patterns that are independent of already well-known disparities for small business and non-RPG research grants.

## Results

### Distribution of funding to RPG PIs over time

*Figure 1* shows different measures of funding distribution to RPG PI's between fiscal years 1985 to 2020. These measures reflect different approaches that economists use to assess income inequality; here we use RPG funding as the analogue of income. We use three different measures:

- Proportion of funds going to the top 1%, or centile, as well as to the top 10%, or decile, (*Saez and Zucman, 2020*) in contrast to the proportion to the bottom 50% (Panel A) or considered alone (Panel C).
- Standard deviation of the log of funding (*Hoffmann et al., 2020*), a measure that accounts for the well-documented skewness in funding and that is particularly sensitive to low and intermediate levels of funding (Panel B).
- The Theil T index (*Conceição and Ferreira, 2000*), a measure that is more sensitive to higher levels of funding (Panel D). Unlike other measures of inequality, the Theil Index is not intuitive. However, it can be used to parse group data, allowing us to parse inequality into *within group* and *between group* components; for example, we can see whether there is a greater degree of inequality between men and women as opposed to within cohorts of men and women.

All three measures indicate greater inequalities in funding since the early 1990s through 2006 corresponding to the NIH-doubling and its aftermath; a plateau from 2006 to 2013; a rapid rise after 2013 (the year of sequestration) to 2017; and a decline approaching 2013 levels from 2018 to 2020. The inequalities are more striking among the most highly funded investigators (Panels C and D), where increases are noted with the NIH doubling (1998 to 2003) and in the first few years after the 2013 budget sequestration. The top 1% of investigators received 8% of RPG funds in 1998; in recent years, they received close to 10% of funds. While this may not seem like much, we should keep in mind that a difference of 2% of RPG funds means that a small group of ~300 investigators are receiving in 2020 approximately $420 million (inflation-adjusted) *more* than they would have received by 1998 standards. Given that the average RPG costs about $500,000, this difference is the equivalent of 800 grants. Inequalities among investigators receiving low to intermediate levels of funding followed a somewhat different trajectory, decreasing during the NIH doubling while increasing after 2013.

### Characteristics of the most highly funded RPG principal investigators

*Table 1* shows characteristics of 34,936 principal investigators funded in fiscal year 2020 according to whether or not they were among the top funded centile. We defined proxies for career stage according to age, with values of 'early' (age < 46), 'middle' (age 46 to 58), and 'late' (age > 58). Compared to the bottom 99%, the top 1% of investigators were in later career stages and more likely to be white, non-Hispanic, and to hold an MD degree (either alone or with a PhD). The difference in funding levels is striking, with top 1% investigators receiving a median of $4.8 million compared to $0.4 million for all others; they were also much more likely to be supported on multiple RPG grants.

*Table 2* shows corresponding characteristics of 19,221 principal investigators funded in fiscal year 1995, before the begining of the NIH doubling. In contrast to 2020, career stage and race differences were less marked, but gender differences were more so. During both eras (before the doubling and in most recent times) top centile investigators were much more likely to hold an MD degree. Consistent with prior literature (*Blau and Weinberg, 2017*), the age range of all NIH funded investigators is skewing older over time. Another noteworthy difference between FY2020 and FY1995 is that much greater proportions of investigators were supported on multiple – 3, 4, or 5 or more – grants in FY2020 than in FY1995.

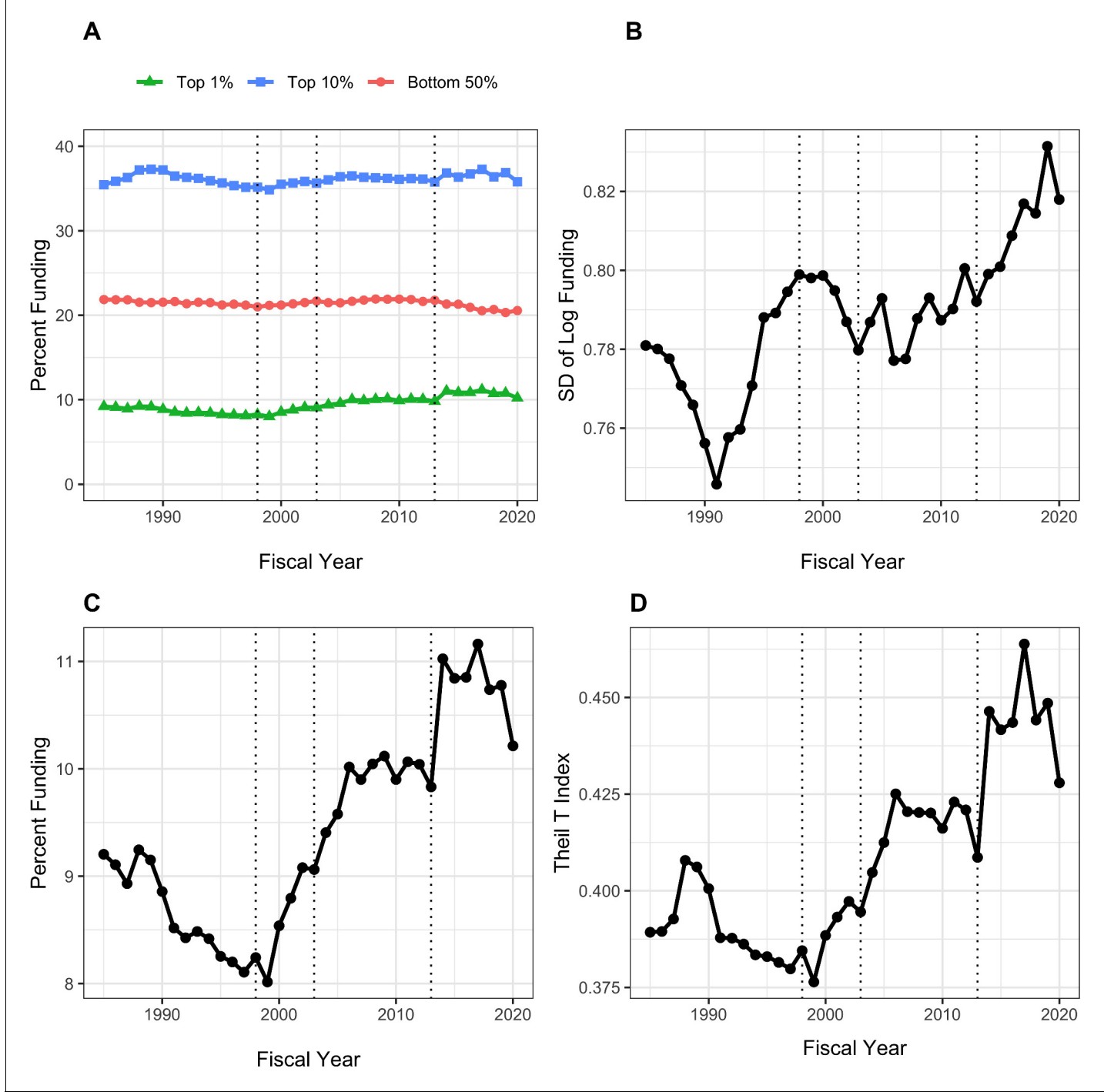

**Figure 1.** Distribution of Research Project Grant (RPG) Principal Investigator (PI) Funding, Fiscal Years 1985–2020. Panel **A**: Percent of RPG funds distributed to the top centile, top decile, and bottom half of investigators. Panel **B**: Standard deviation of the log of funding, a measure that focuses primarily on lower and intermediate levels of funding. Panel **C**: Percent of RPG funds distributed solely to the top centile of investigators. Panel **D**: Theil T index, a measure more sensitive to the highest funding levels, and hence has a similar appearance to percent of funds distributed to the top centile. The vertical dotted lines refer to the beginning and end of the NIH doubling and the year of budget sequestration (2013).

## Inequalities between and within groups

*Figure 2* shows secular changes in composition of the RPG PI workforce between FY1985 and FY2020. Over time, there have been increases in the proportion of late career, female, and Asian

**Table 1.** Investigator characteristics according to centile of funding in fiscal year 2020.
Values shown in parentheses are percentages for categorical variables and IQR for continuous variables. IQR = inter-quartile range. ND = not displayed due to small cell size.

| Characteristic | | Top 1% | Bottom 99% |
|---|---|---|---|
| Total N (%) | | 349 (1.0) | 34587 (99.0) |
| Career Stage | Early | 30 (8.6) | 10567 (30.6) |
| | Middle | 128 (36.7) | 12936 (37.4) |
| | Late | 162 (46.4) | 8273 (23.9) |
| Gender | Female | 102 (29.2) | 11858 (34.3) |
| | Male | 241 (69.1) | 21695 (62.7) |
| Race | White | 277 (79.4) | 23264 (67.3) |
| | Asian | 42 (12.0) | 7523 (21.8) |
| | Black or African-American | ND | 639 (1.8) |
| | More than One Race | ND | 418 (1.2) |
| Ethnicity | Hispanic | 12 (3.4) | 1622 (4.7) |
| | Not Hispanic | 306 (87.7) | 29513 (85.3) |
| Degree | PhD | 166 (47.6) | 24620 (71.2) |
| | MD | 116 (33.2) | 5238 (15.1) |
| | MD-PhD | 60 (17.2) | 3572 (10.3) |
| | Other | ND | 1157 (3.3) |
| Funding in $Million | Median (IQR) | 4.8 (4.0 to 6.5) | 0.4 (0.3 to 0.7) |
| Number of RPG Awards | One | 69 (19.8) | 23268 (67.3) |
| | Two | 86 (24.6) | 7571 (21.9) |
| | Three | 52 (14.9) | 2540 (7.3) |
| | Four | 60 (17.2) | 847 (2.4) |
| | Five or More | 82 (23.5) | 361 (1.0) |

investigators. Middle career investigators are comprising a lower proportion of the workforce since the mid-2000s. Over the past 4–5 years, the proportions of PIs at different career stages have stabilized. The proportion of late-career investigators is no longer rising while that of mid-career investigators is no longer falling. This stabilization has occured at the same as NIH implementation of its Next Generation Researchers Initiative (*Lauer et al., 2017*). The fraction of women among funded PIs continues to increase, but they are still not at parity. The proportion of MD-only degree holders has fallen, while the proportion of MD-PhD degree holders has increased. *Figure 3* shows using box plots the FY 2020 distribution of funding to RPG PIs according to career stage, gender, race, and degree. Late career investigators, men, whites, and those holding MD degrees are better funded. Nonetheless, one notes that there appears to be greater variability within groups than between groups.

Table 3 shows FY2020 characteristics according to career stage. Late-career investigators were more likely to be white males, to hold MD degrees, and to be designated as PI on a larger number of grants. *Table 4* shows FY2020 investigator characteristics according to gender. Women were younger, more likely to hold a PhD degree, and less likely to be principal investigators of 2 or more RPG grants. *Table 5* shows corresponding race data. Black or African-American investigators were younger, more likely to be women, and more likely to hold MD degrees. They were also much more likely to serve a PI on only one RPG grant.

The Theil T index enables us to formally assess between-group and within-group contributions to inequality. *Figure 4* shows that for all groupings, within group differences contribute more to inequality than between-group differences. The small between-group differences are shown in *Figure 5*. Late stage investigators, men, whites, and investigators with MD degrees contribute 'positive

**Table 2.** Investigator characteristics according to centile of funding in fiscal year 1995.
Data on ethnicity are not provided due to high rates of missingness (more than one-third). Dollar values are inflation-adjusted to a FY2019 reference standard. Values shown in parentheses are percentages for categorical variables and IQR for continuous variables. IQR = inter-quartile range. ND = not displayed due to small cell size.

| Characteristic | | Top 1% | Bottom 99% |
|---|---|---|---|
| Total N (%) | | 192 (1.0) | 19029 (99.0) |
| Career Stage | Early | 37 (19.3) | 8757 (46.0) |
| | Middle | 111 (57.8) | 7305 (38.4) |
| | Late | 33 (17.2) | 1852 (9.7) |
| Gender | Female | 23 (12.0) | 4266 (22.4) |
| | Male | 165 (85.9) | 14439 (75.9) |
| Race | White | 167 (87.0) | 16121 (84.7) |
| | Asian | 14 (7.3) | 1525 (8.0) |
| | Black or African-American | ND | 164 (0.9) |
| | More than One Race | ND | 89 (0.5) |
| Degree | PhD | 72 (37.5) | 13418 (70.5) |
| | MD | 81 (42.2) | 3740 (19.7) |
| | MD-PhD | 38 (19.8) | 1703 (8.9) |
| | Other | ND | 168 (0.9) |
| Funding in $Million | Median (IQR) | 4.5 (4.0 to 5.7) | 0.4 (0.3 to 0.7) |
| Number of RPG Awards | One | 46 (24.0) | 14894 (78.3) |
| | Two | 61 (31.8) | 3391 (17.8) |
| | Three | 57 (29.7) | 617 (3.2) |
| | Four | 22 (11.5) | 115 (0.6) |
| | Five or More | ND | 12 (0.1) |

elements' because they on average receive higher levels of funding. Nonetheless, the absolute values of these elements, as compared to the total Theil index, are small.

## Organizational inequalities

In additional analyses, we look at RPG funding inequalities among organizations. *Figure 6* shows data analagous to those in *Figure 1*. Because the absolute number of organizations is much less than for PIs (e.g. in 2020 there were 1097 unique organizations receiving RPG funding) we focus on the top decile (10%) rather than the top centile. The top 10% of organizations have been receiving approximately 70% of RPG funding, while the bottom half have received well under 5%. Like with PIs, inequalities increased after the doubling, but patterns in more recent years have differed. Inequalities decreased in the late 2000s (perhaps coincident with the 2008 finanical crash), but have increased slightly in more recent years.

*Figure 7* shows distribution of RPG funding in Fiscal Year 2020 according to organization type. Because the distributions are highly skewed (even more so than with PIs), we show log-transformed values (Panel A). There are marked differences *between* groups – medical schools are receiving higher levels of funding than other institutions. We confirm this by calculating Theil indices, which show that organizational inequalities stem from both *between group* and *within group* variability (Panel B). The Theil elements plot (Panel C), consistent with Panel A, shows that medical schools, and to a lesser extent hospitals, are groups that receive higher levels of funding. *Figure 8* shows corresponding data according to organization region. Funding inequalities were greater within regions than between regions. *Figure 9* shows similarly that for domestic institutions within state inequalities contribute more to overall inequality that between-state inequalities.

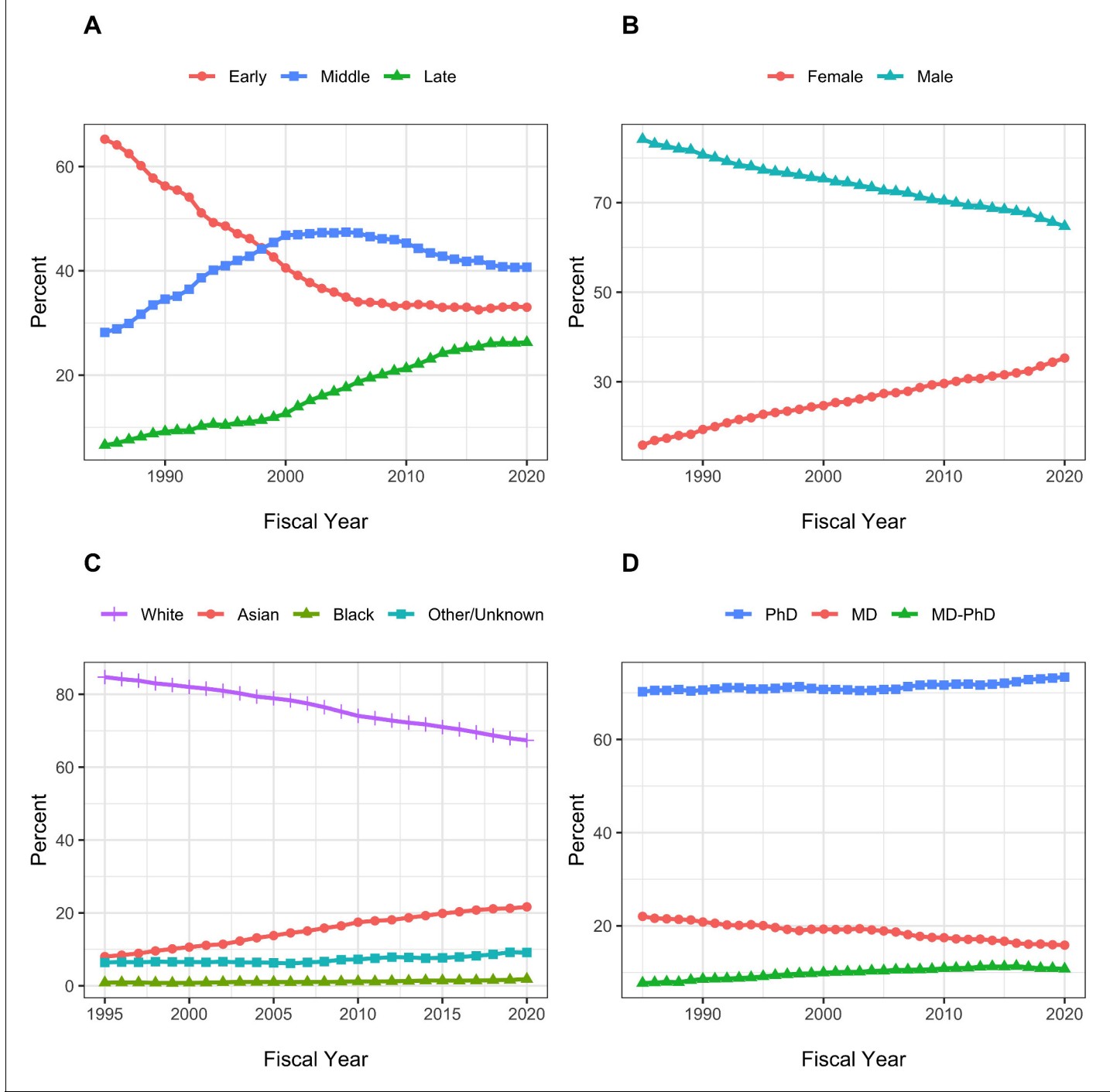

**Figure 2.** Secular changes in the composition of the RPG PI Workforce from fiscal year 1985 to fiscal year 2020. Race data are shown from 1995 on due to high proportions of unknown values beforehand. Each plot shows the percentage of RPG PIs according to different groupings. All percentages add up to 100. Panel **A**: Career Stage. Panel **B**: Gender. Panel **C**: Race. Panel **D**: Degree.

## Perspective: Income inequality in the united States and Europe – Population Data

In order to put these NIH-specific data into perspective, we present high-level income equality data for general populations of the United States and the European Union. We show data from the World Inequality Database (**Saez, 2021**), which was developed by Emmanuel Saez and colleagues.

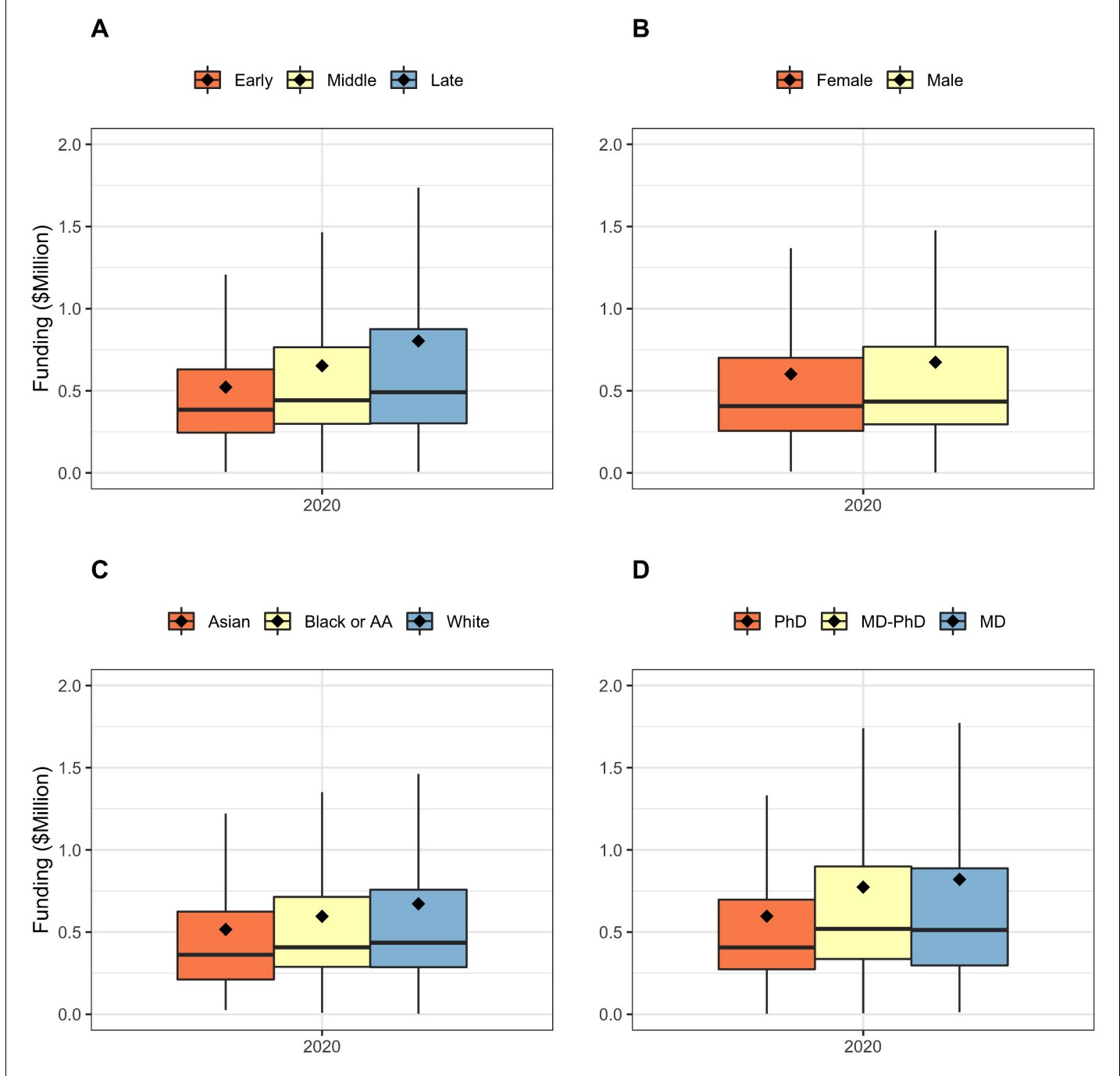

**Figure 3.** Box plots showing the distribution of funding in FY2020 according to PI groups. Diamonds refer to means; the higher means compared to medians reflect highly skewed distributions. Outliers are not displayed Panel **A**: Career Stage. Panel **B**: Gender. Panel **C**: Race. Panel **D**: Degree. For all groups, variability appears to be greater within groups than between groups. AA = African-American.

*Figure 10* shows percent of annual income going to the top centile (Panel A) and the bottom half (Panel B) of the populations of the United States and Europe from 1980 to 2020. We focus on income, instead of wealth, since income for most people comes from remuneration for work and therefore would be analogous to RPG funding awarded in anticipation of scientific work. At all times, income inequality has been greater in the United States. Changes in inequality have also been greater in the United States. From 1995 to 2019, the proportion of income going to the top centile of the United States population has increased from 14.3% to 18.7%, a relative increase of 31%.

**Table 3.** Investigator characteristics according to career stage in fiscal year 2020.
Values shown in parentheses are percentages for categorical variables and IQR for continuous variables. IQR = inter-quartile range.

| Characteristic | | Early | Middle | Late |
| --- | --- | --- | --- | --- |
| Total N (%) | | 10597 (30.3) | 13064 (37.4) | 8435 (24.1) |
| Gender | Female | 4241 (40.0) | 4505 (34.5) | 2267 (26.9) |
| | Male | 6145 (58.0) | 8464 (64.8) | 6128 (72.6) |
| Race | White | 6855 (64.7) | 8509 (65.1) | 6990 (82.9) |
| | Asian | 2515 (23.7) | 3440 (26.3) | 955 (11.3) |
| | Black or African-American | 270 (2.5) | 232 (1.8) | 84 (1.0) |
| | More than One Race | 209 (2.0) | 153 (1.2) | 42 (0.5) |
| Degree | PhD | 8643 (81.6) | 9115 (69.8) | 5355 (63.5) |
| | MD | 1067 (10.1) | 2006 (15.4) | 1935 (22.9) |
| | MD-PhD | 651 (6.1) | 1714 (13.1) | 998 (11.8) |
| | Other | 236 (2.2) | 229 (1.8) | 147 (1.7) |
| Funding in $Million | Median (IQR) | 0.4 (0.2 to 0.6) | 0.4 (0.3 to 0.8) | 0.5 (0.3 to 0.9) |
| Funding Percentile Rank | Median (IQR) | 55.3 (31.7 to 78.2) | 47.2 (23.2 to 72.2) | 42.5 (18.6 to 71.7) |
| Number of RPG Awards | One | 7704 (72.7) | 8149 (62.4) | 5377 (63.7) |
| | Two | 2047 (19.3) | 3163 (24.2) | 1957 (23.2) |
| | Three | 584 (5.5) | 1155 (8.8) | 701 (8.3) |
| | Four | 183 (1.7) | 384 (2.9) | 268 (3.2) |
| | Five or More | 79 (0.7) | 213 (1.6) | 132 (1.6) |

**Table 4.** Investigator characteristics according to gender in fiscal year 2020.
Values shown in parentheses are percentages for categorical variables and IQR for continuous variables. IQR = inter-quartile range.

| Characteristic | | Women | Men |
| --- | --- | --- | --- |
| Total N (%) | | 11960 (34.2) | 21936 (62.8) |
| Career Stage | Early | 4241 (35.5) | 6145 (28.0) |
| | Middle | 4505 (37.7) | 8464 (38.6) |
| | Late | 2267 (19.0) | 6128 (27.9) |
| Race | White | 8528 (71.3) | 14876 (67.8) |
| | Asian | 2405 (20.1) | 5106 (23.3) |
| | Black or African-American | 296 (2.5) | 342 (1.6) |
| | More than One Race | 189 (1.6) | 230 (1.0) |
| Degree | PhD | 9093 (76.0) | 15278 (69.6) |
| | MD | 1734 (14.5) | 3540 (16.1) |
| | MD-PhD | 867 (7.2) | 2732 (12.5) |
| | Other | 266 (2.2) | 386 (1.8) |
| Funding in $Million | Median (IQR) | 0.4 (0.3 to 0.7) | 0.4 (0.3 to 0.8) |
| Funding Percentile Rank | Median (IQR) | 51.3 (27.1 to 76.7) | 48.0 (23.1 to 72.7) |
| Number of RPG Awards | One | 8409 (70.3) | 14002 (63.8) |
| | Two | 2512 (21.0) | 5066 (23.1) |
| | Three | 732 (6.1) | 1833 (8.4) |
| | Four | 212 (1.8) | 688 (3.1) |
| | Five or More | 95 (0.8) | 347 (1.6) |

**Table 5.** Investigator characteristics according to race in fiscal year 2020.

Values shown in parentheses are percentages for categorical variables and IQR for continuous variables. IQR = inter-quartile range. ND = not displayed due to small cell size.

| Characteristic | | White | Asian | Black or African-American |
|---|---|---|---|---|
| Total N (%) | | 23541 (67.4) | 7565 (21.7) | 643 (1.8) |
| Career Stage | Early | 6855 (29.1) | 2515 (33.2) | 270 (42.0) |
| | Middle | 8509 (36.1) | 3440 (45.5) | 232 (36.1) |
| | Late | 6990 (29.7) | 955 (12.6) | 84 (13.1) |
| Gender | Female | 8528 (36.2) | 2405 (31.8) | 296 (46.0) |
| | Male | 14876 (63.2) | 5106 (67.5) | 342 (53.2) |
| Degree | PhD | 17086 (72.6) | 5398 (71.4) | 406 (63.1) |
| | MD | 3831 (16.3) | 976 (12.9) | 141 (21.9) |
| | MD-PhD | 2211 (9.4) | 1094 (14.5) | 73 (11.4) |
| | Other | 413 (1.8) | 97 (1.3) | 23 (3.6) |
| Funding in $Million | Median (IQR) | 0.4 (0.3 to 0.8) | 0.4 (0.3 to 0.7) | 0.4 (0.2 to 0.6) |
| Funding Percentile Rank | Median (IQR) | 47.8 (23.6 to 73.6) | 51.3 (26.3 to 73.5) | 60.7 (32.1 to 83.8) |
| Number of RPG Awards | One | 15506 (65.9) | 4914 (65.0) | 504 (78.4) |
| | Two | 5344 (22.7) | 1688 (22.3) | 103 (16.0) |
| | Three | 1784 (7.6) | 616 (8.1) | 25 (3.9) |
| | Four | 611 (2.6) | 230 (3.0) | ND |
| | Five or More | 296 (1.3) | 117 (1.5) | ND |

During the same time, the proportion of RPG funding going to the top centile of RPG PIs has increased from 8.3% to 10.8% (*Figure 1*, Panel C), a relative increase of 30%. Although the US population and NIH-funded PIs have experienced different events – e.g., the 2000 recession and the 2008 financial crash for the US population; the NIH doubling, the 2006 payline crash, the 2013 sequestration, and the recent string of budget increases for NIH-funded PIs – the overall relative changes in inequality at the top are remarkably similar.

## Discussion

Inequalities in funding of RPG PIs have increased since the NIH doubling, with further increases since sequestration in 2013 (*Figure 1*). Over the past few years, a time of substantial and sustained budget increases for NIH and a time of focus on early career investigators, there has been a decrease in the degree of inequality, but not quite back to the level of 2013. The RPG funding inequalities primarily reflect changes 'at the top,' meaning among the most highly funded investigators (*Figure 1*, Panels C and D). The top 1%'s share of RPG funding has increased from 8% before the doubling to nearly 10% now (*Figure 1*, Panel C); this difference translates into ~$400 million, or the equivalent of 800 RPG awards. Since sequestration, the top 1% has received an increased share of funding, while the bottom 50% has received less. During the NIH doubling, both the top centile and the lower half saw increases in the proportion of funding they received (*Figure 1*, Panels A and B).

The composition of the RPG PI workforce has evolved over time, with greater proportions of investigators who are late career, women, and Asian, and lesser proportions of MD-only degree holders (*Figure 2*). Despite steady increases in the proportion of women investigators, they are still well below parity. (*Figure 2*, Panel B). Among the groups studied, more funding goes to late career investigators, as well as to men, whites, and holders of MD degrees. Nonetheless, there is greater inequality within groups than between groups (*Figures 3–5*). One might argue that it may be reasonable for researchers to receive more funding at later career stages as they may have larger networks and are more experienced at posing research questions. Thus, some inequality may be considered 'acceptable.' But there is not funding parity for gender or race for researchers in the workforce, which are unacceptable inequalities. Over the past few years NIH has launched high-

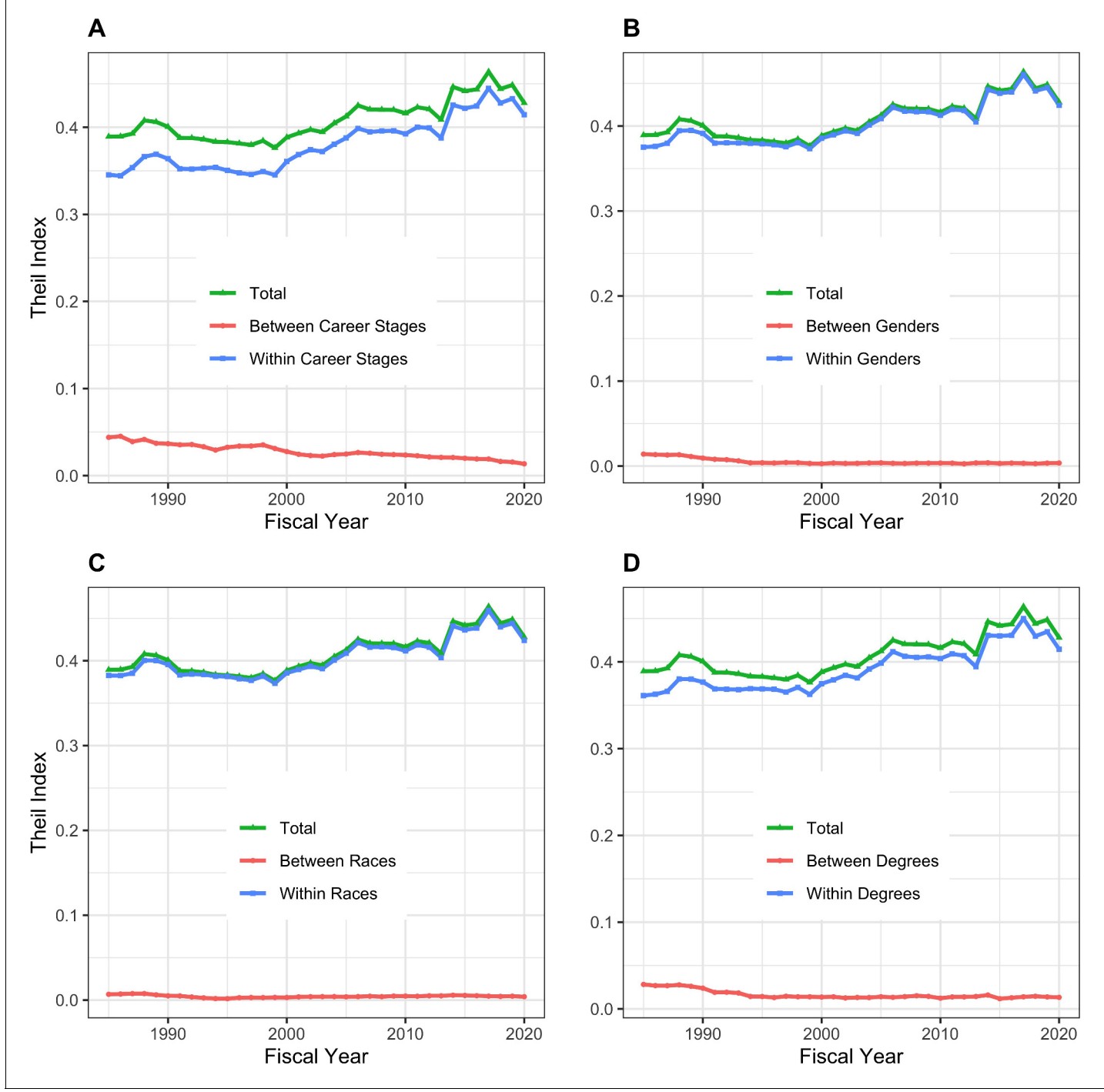

**Figure 4.** Components of Theil index, showing between-group and within-group contributions to overall inequality over time. Panel **A**: Career Stage. Panel **B**: Gender. Panel **C**: Race. Panel **D**: Degree. For all groups, within-group differences contribute more to inequality than between-group differences.

profile initiatives to enhance the diversity of the biomedical research work force. (*National Institutes of Health, 2021b*; *ORWH, 2021*).

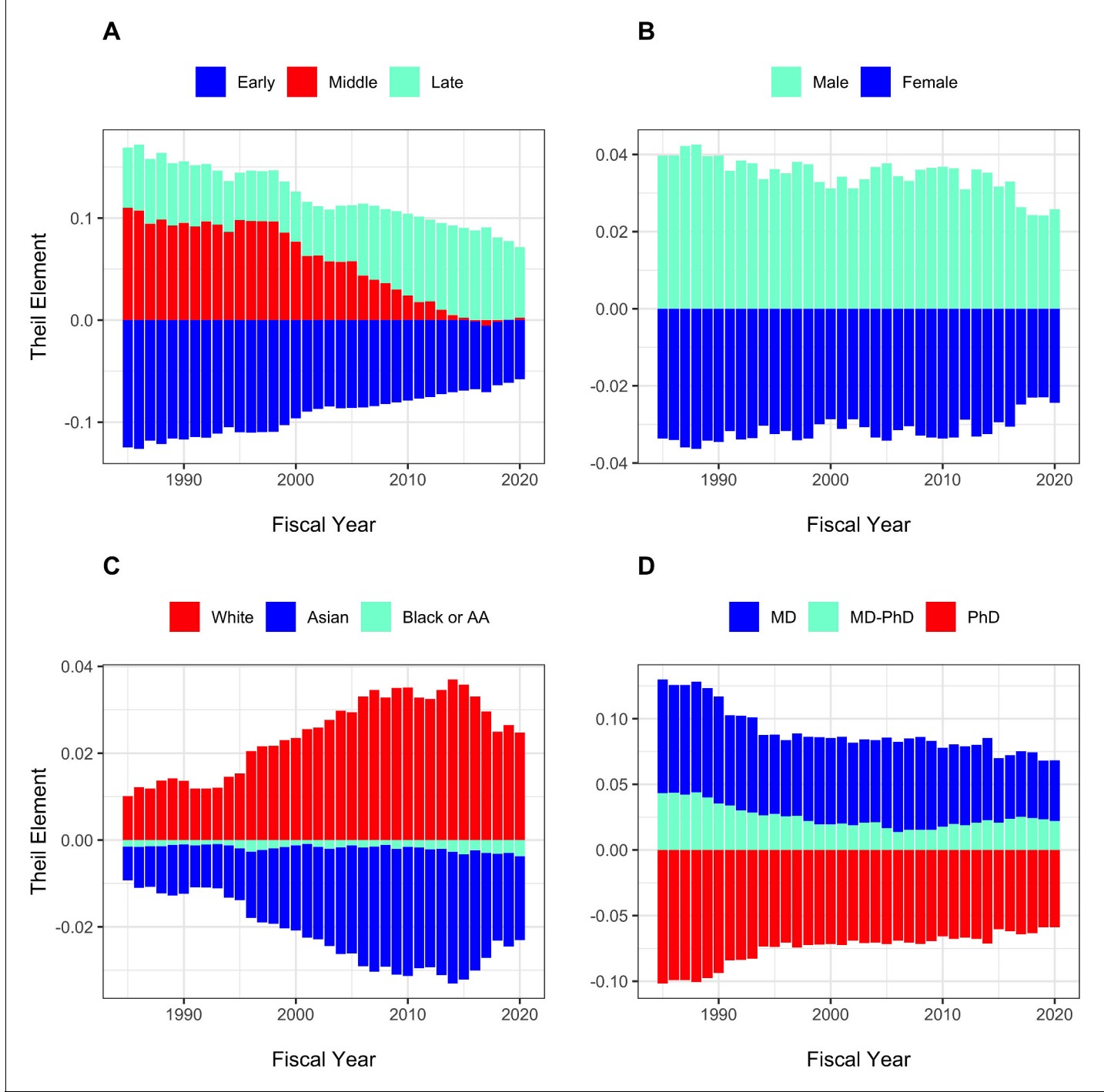

**Figure 5.** Theil Elements in different groups over time. Panel **A**: Career stage. Panel **B**: Gender. Panel **C**: Race. Panel **D**: Degree. Values above the zero line indicate that groups received above average funding, while values below zero indicate below average funding. Thus, as in Panel A, late stage investigators received above average funding and early stage investigators received below average funding. Middle career investigators initially received above average funding, but in recent years have received funding close to average, contributing little to inequality. AA = African-American.

## Materials and methods

From the NIH IMPAC II database, we obtained PI-specific data on inflation-adjusted total-cost funding of Research Project Grants (RPGs), defined as those grants with activity codes of DP1, DP2, DP3, DP4, DP5, P01, PN1, PM1, R00, R01, R03, R15, R21, R22, R23, R29, R33, R34, R35, R36, R37, R61,

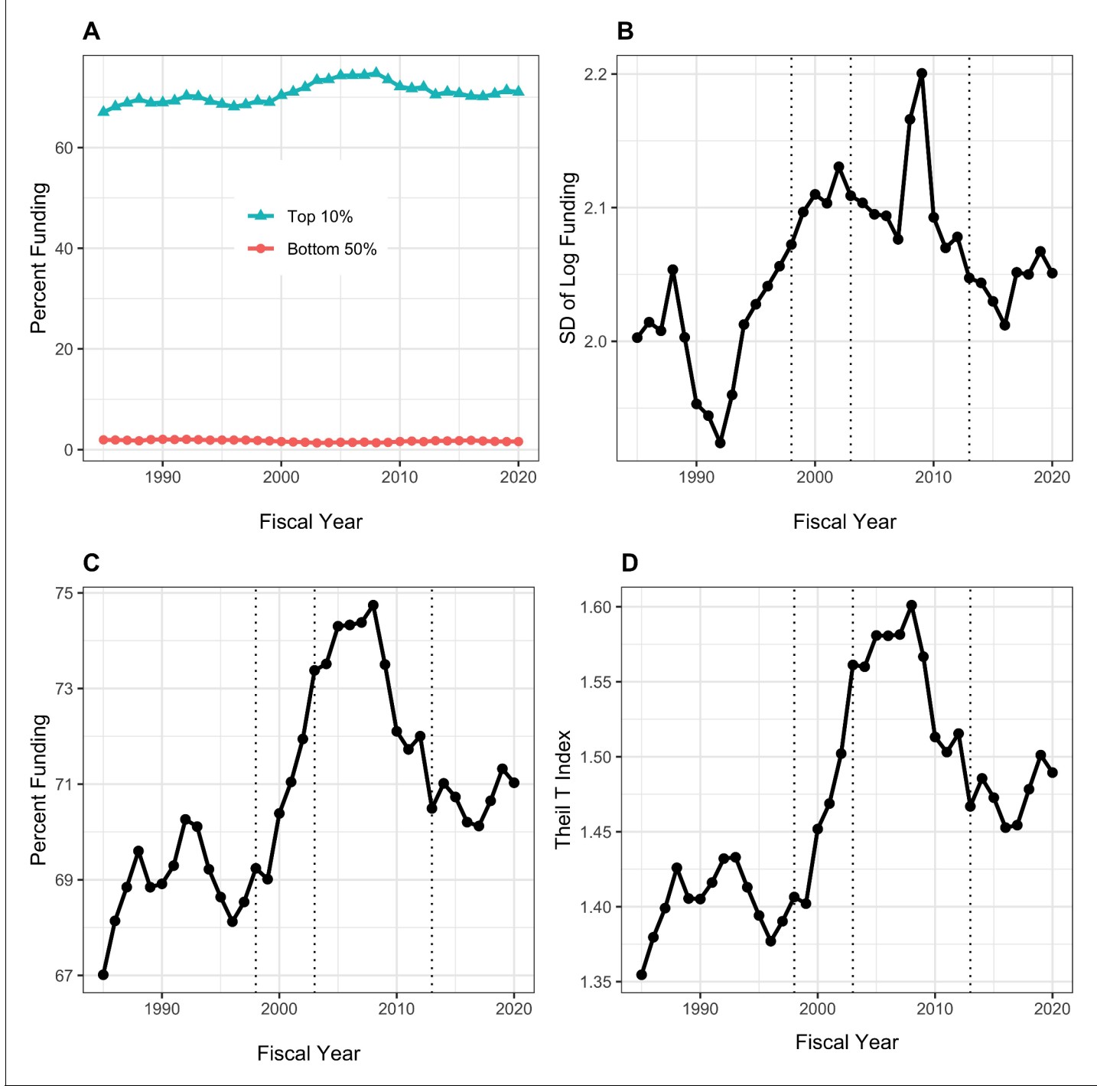

**Figure 6.** Distribution of Research Project Grant (RPG) Organization Funding, Fiscal Years 1985–2020. Panel **A**: Percent of RPG funds distributed to the top decile and bottom half of organizations. Panel **B**: Standard deviation of the log of funding, a measure that focuses primarily on lower and intermediate levels. Panel **C**: Percent of RPG funds distributed solely to the top decile of organizations. Panel **D**: Theil T index, a measure more sensitive to the highest funding levels, and hence has a similar appearance to percent of funds distributed to the top centile. The vertical dotted lines in Panels B, C, and D refer to the beginning and end of the NIH doubling and the year of budget sequestration (2013).

R50, R55, R56, RC1, RC2, RC3, RC4, RF1, RL1, RL2, RL9, RM1, UA5, UC1, UC2, UC3, UC4, UC7, UF1, UG3, UH2, UH3, UH5, UM1, UM2, U01, U19, and U34. Not all of these activity codes were used by NIH every year. For FY 2009 and 2010, we excluded awards made under the American Recovery and Reinvestment Act of 2009 (ARRA) and all ARRA solicited applications and awards. For

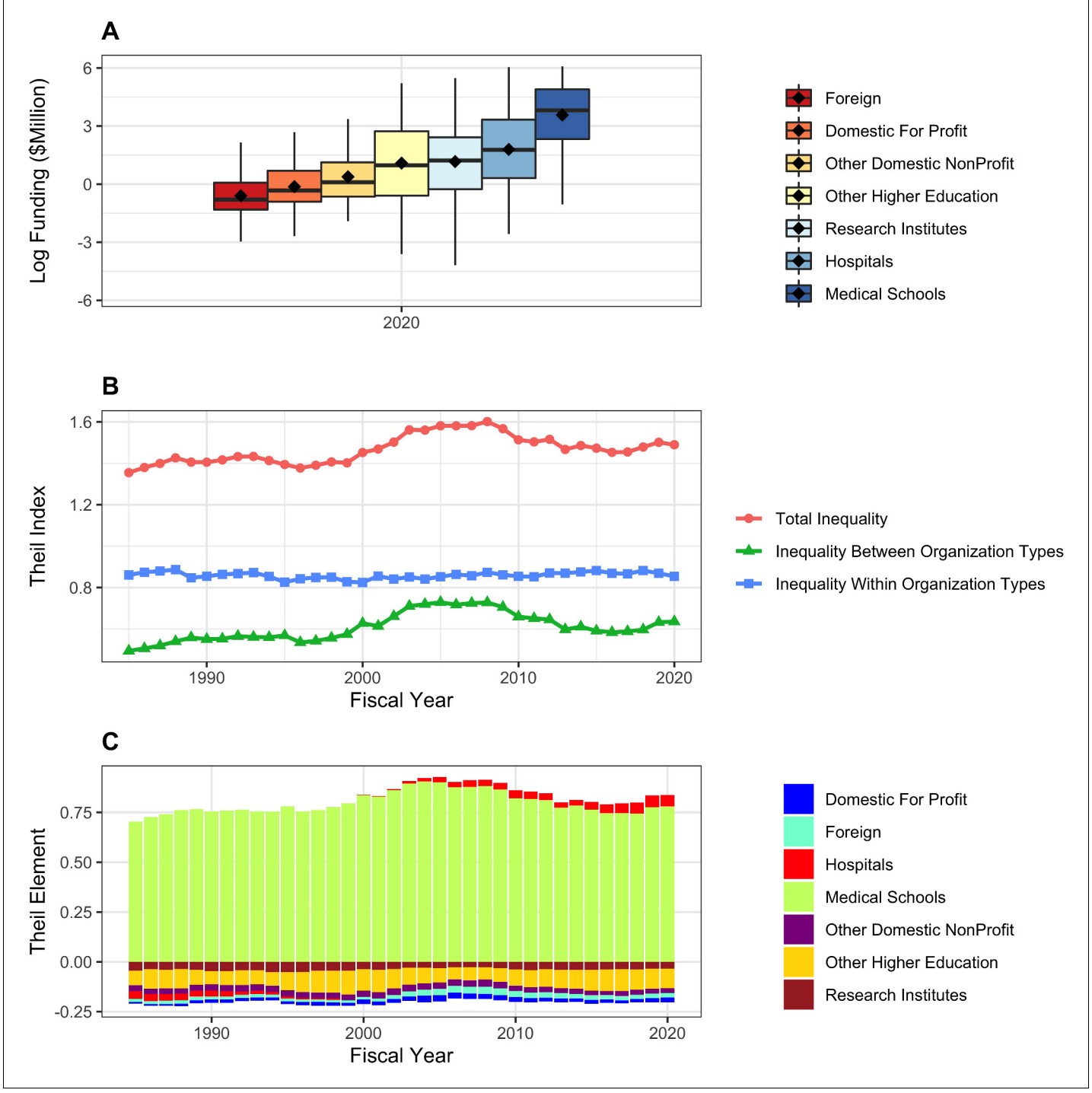

**Figure 7.** RPG funding distribution and inequalities according to organization type. Panel **A**: Box plots showing distributions of log-transformed RPG funding in FY2020. Panel **B**: Theil index components plot, showing that both between group and within group inequalities contribute to overall inequality. Panel **C**: Theil elements plot. Values above the zero line indicate that groups received above average fundings, while values below zero indicate below average funding. Medical schools and hospitals received above average funding.

FY2020, we excluded awards issued using supplemental Coronavirus (COVID-19) appropriations. Inflation-adjustments were referenced to FY2019 using the Biomedical Research and Development Price Index (*National Institutes of Health, 2021a*).

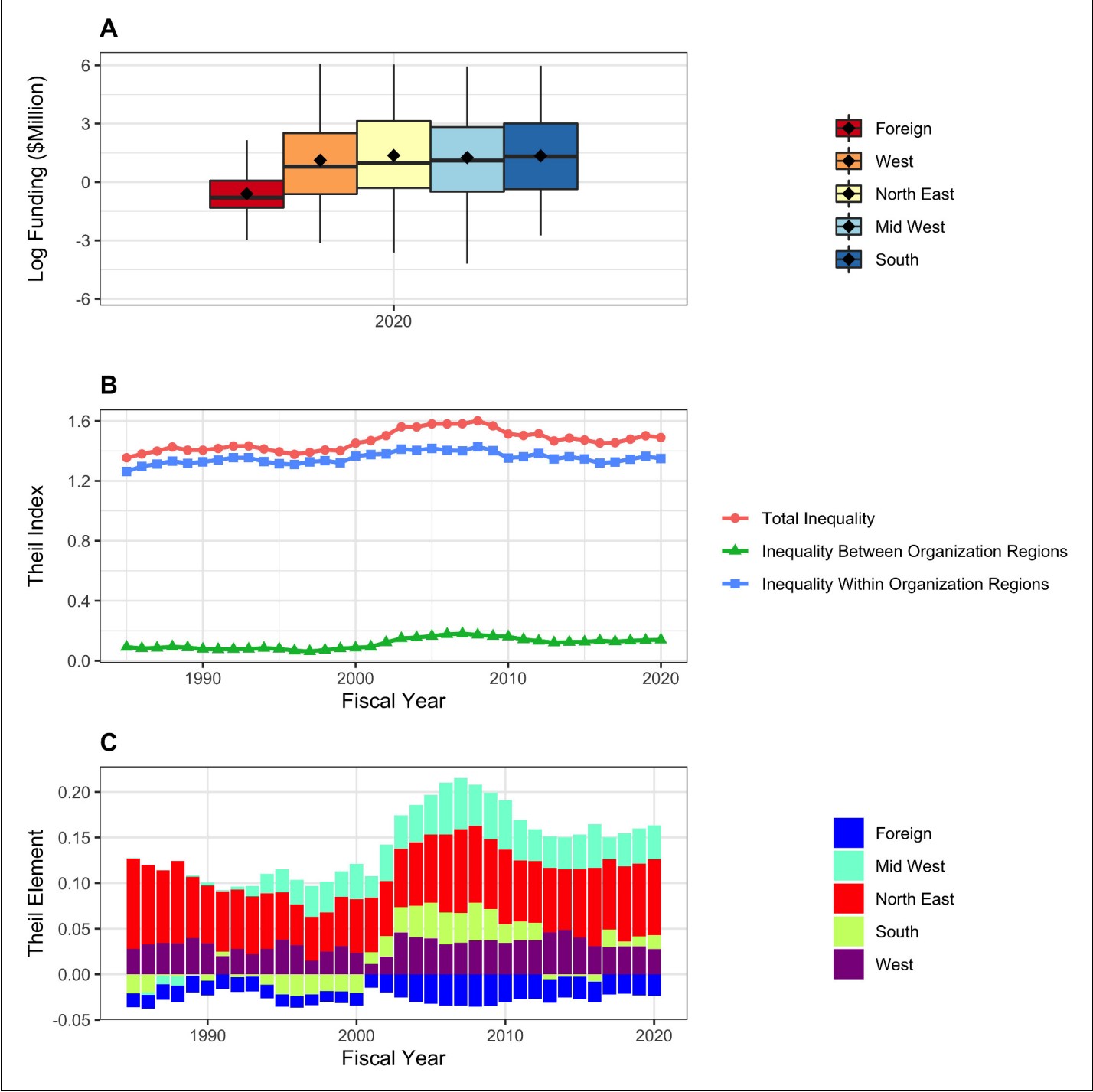

**Figure 8.** RPG funding distribution and inequalities according to organization region. Panel **A**: Box plots showing distributions of log-transformed RPG funding in FY2020. Panel **B**: Theil index components plot, showing that within group inequalities primarily contribute to overall inequality. Panel **C**: Theil elements plot. Values above the zero line indicate that groups received above average fundings, while values below zero indicate below average funding. Foreign organizations received below average funding.

We measured inequality by three approaches: Proportion of funds going to the top 1%, or centile, (*Saez and Zucman, 2020*); standard deviation of the log of funding (*Hoffmann et al., 2020*), a measure that accounts for the well-documented skewness in funding and that is particularly sensitive to low and intermediate levels of funding; and the Theil T index (*Conceição and Ferreira, 2000*), a

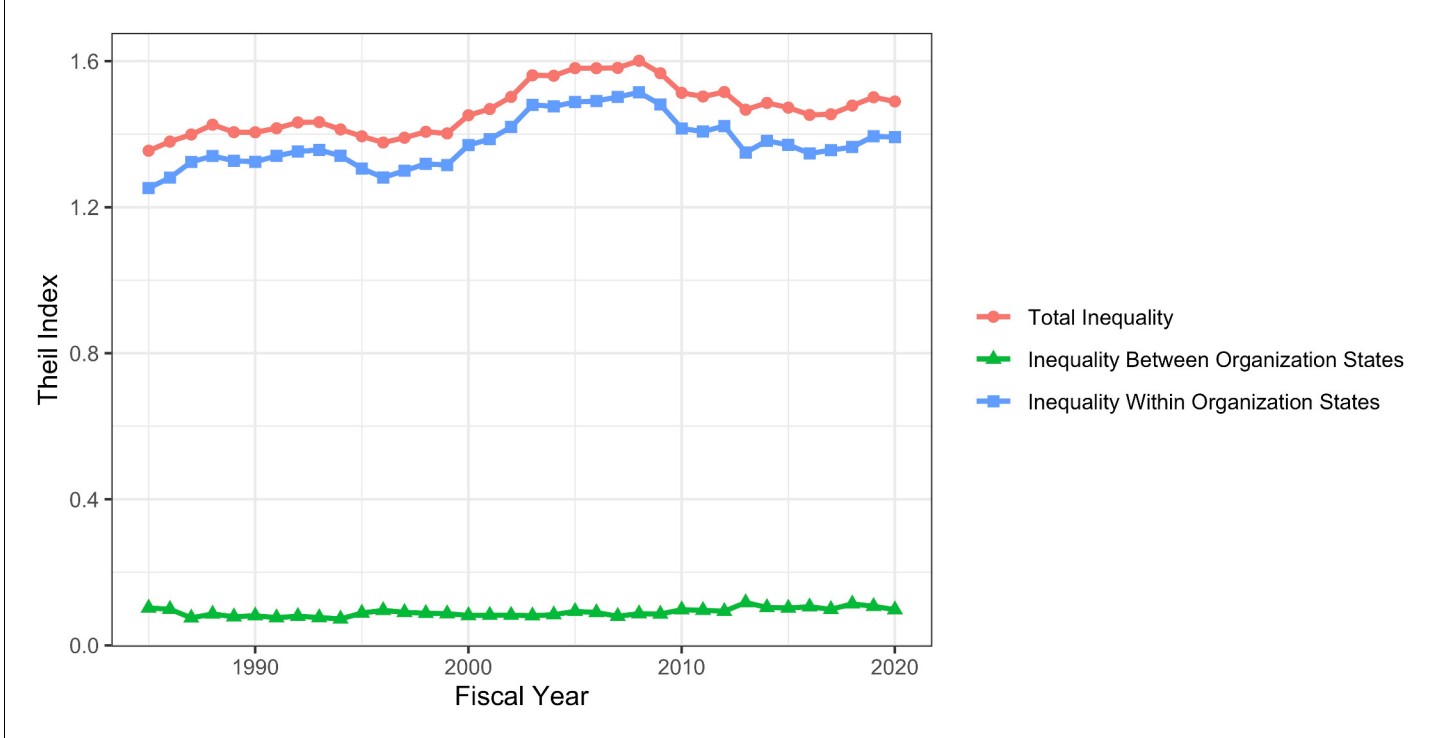

**Figure 9.** RPG funding distribution and inequalities according to organization state within the United States. The panel shows a Theil index components plot, showing that within state inequalities contribute more to overall inequality than between-state inequality.

measure that is more sensitive to higher levels of funding and that be exploited to explore contributions of different groups to overall inequality.

For individual level data (say of individual PIs), the Theil Index (T) of funding inequality is mathematially represented as:

$$T = \sum_{p=1}^{n}(\frac{1}{n} * \frac{y_p}{\mu_y} * ln\frac{y_p}{\mu_y}) \tag{1}$$

where $n$ is the number of individual PIs, $y_p$ is the funding of PI $p$, and $\mu_y$ is the population mean funding. The final logarithmic fraction takes on a value greater than 0 if the individual investigator $p$'s funding is greater than the population mean $\mu_y$ and less than 0 if the individual investigator's funding is less than the population mean. We can think of the three terms as: $\frac{1}{n}$ as the investigator's proportion of the population; $\frac{y_p}{\mu_y}$ as the magnitude of deviance compared to the overall population; and $ln\frac{y_p}{\mu_y}$ as the direction of deviance.

For grouped data (e.g. data grouped by career stage or gender or other characteristics), we can present the Theil Index $T$ as a weighted average of inequality *within* each group plus inequality *between* those groups. That is:

$$T = T'_g + T^w_g \tag{2}$$

where $T'_g$ is the *between-group* component and $T^w_g$ is the *within group* component.

The *between-group* component of the Theil Index ($T'_g$) is mathematically represented in a form similar to the overall Theil Index (*Equation 1*), namely:

$$T'_g = \sum_{i=1}^{m}(\frac{p_i}{P} * \frac{y_i}{\mu} * ln\frac{y_i}{\mu}) \tag{3}$$

where $i$ indexes the $m$ groups (e.g. early, middle, and late career investigators), $P$ is the total

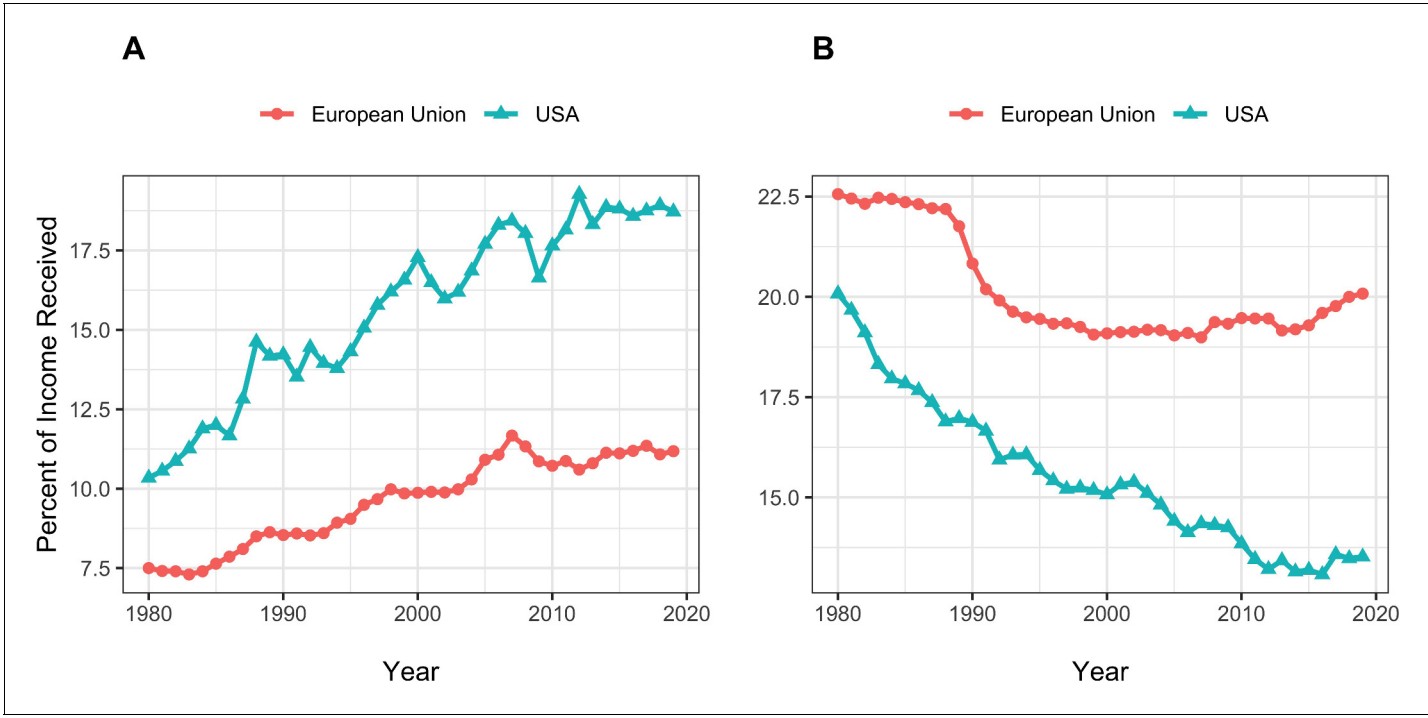

**Figure 10.** United States and European Union income equality measures from the World Inequality Database. Panel **A**: Percent of income going to the top centile of the population. Panel **B**: Percent of income going to the bottom half.

The online version of this article includes the following source data for figure 10:

**Source data 1.** Data from the World Inequality Database.

population, $y_i$ is the average funding of the group $i$, and $\mu$ is the average funding accross the entire population. The expression within the parenthesis is called the 'Theil element,' which is positive (or negative) if the group's average funding is above (or below) the population average and zero if the averages are equal. The Theil elements represent the contribution of each group to total inequality *between* the groups.

Unlike other measures of inequality (e.g. proportion of funding going to the top centile or standard deviation of log funding), the Theil Index is not intuitive. However, it can be used to parse group data, allowing us to parse inequality into *within group* and *between group* componentd *between group* component.

## Additional information

### Funding

| Funder | Author |
| --- | --- |
| National Institutes of Health | Michael S Lauer<br>Deepshikha Roychowdhury |

The funders had no role in study design, data collection and interpretation, or the decision to submit the work for publication.

### Author contributions

Michael S Lauer, Conceptualization, Formal analysis, Supervision, Methodology, Writing - original draft; Deepshikha Roychowdhury, Conceptualization, Data curation, Formal analysis, Methodology, Writing - review and editing

**Author ORCIDs**

Michael S Lauer [ID] https://orcid.org/0000-0002-9217-8177

**Decision letter and Author response**

Decision letter https://doi.org/10.7554/eLife.71712.sa1
Author response https://doi.org/10.7554/eLife.71712.sa2

## Additional files

**Supplementary files**

- Source code 1. Source code for entire paper.
- Source data 1. Anonymized organization data.
- Source data 2. Anonymized PI data.
- Transparent reporting form

**Data availability**

Source data have been provided in R format. R markdown source code corresponds with all numbers, tables, and figures.

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
