## [Decision Letter]

**Acceptance summary:**

This article by Lauer et al. provides new insights into the important problem of inequalities in grant distribution in awards from the NIH. The article highlights that differences in funding rates within a group in the areas of race and gender persist, despite active efforts over the last several years to close these gaps. It is critical to address these inequalities if there is to be continued growth in the pace of scientific discovery.

**Decision letter after peer review:**

Thank you for submitting your article "Inequalities in the Distribution of National Institutes of Health Research Project Grant Funding" for consideration by *eLife*. Your article has been reviewed by 2 peer reviewers, and the evaluation has been overseen by a Senior Editor and Mone Zaidi as the Deputy Editor. The following individual involved in review of your submission has agreed to reveal their identity: Mark Peifer (Reviewer #1).

The reviewers have discussed their reviews with one another, and the Senior Editor has drafted this to help you prepare a revised submission.

Essential revisions:

1. Revised wording/clarifications to make sure conclusions better match the data.

a. Abstract. First, stating the trend toward reversing inequality has "reversed", while technically accurate, implies return to the start of the analysis. In fact, there has been only a modest reversal, and inequality remains very strong. This sentence should be more nuanced, as should the following statement about career stages. Second the statement about women is very confusing. They state "Women continue to constitute a greater proportion of funded principal investigators, though not at parity." When I first read this, I was thinking – no chance that women are "a greater proportion", i.e., a majority of funded PIs. It would be clearer to state: "The fraction of women among funded PIs continues to increase, but they are still not at parity".

b. Figure 1 should have the Y-axis go to zero, to make it clearer just how much funding is held by the top 1%.

c. P. 4. The authors should note that the age range of ALL NIH funded investigators is skewing older over time – this has been analyzed by others but is worth a mention.

d. The data in Figure 2 is very interesting, and deserves a more complete description, especially as it brings these analyses up to the current date.

e. P 10, Figure 3, the choice of data presentation and the way it was discussed significantly minimized the differences by career stage, race and gender. I imagine the majority of investigators in all groups are funded by a single grant at or near the modular level, and this is leveling differences – the mean/median difference supports this. I suspect that, and the authors have data to explore whether, the most highly funded individuals are very different between groups. What would the top decile look like for each group – their mean values suggest substantial differences? I'd like to see more in-depth analysis here. I think the authors have dug deeper, as they state "Women were younger, more likely to hold a PhD degree, and less likely to be principal investigators of more than the equivalent of 4 R01 grants.", but how one would assess this last factor from the data presented is not clear.

f. The analysis of different institutions was stunning (Figures 6 and 7), and deserves some mention in the abstract and some more text talking about the nuances. I also wondered if the authors have analyzed by region of the country, as this is also an area that has attracted attention (e.g. analyses of Walls)?

2. Additional analyses

a. The analysis in Figure 1 was exceptionally interesting. A similar comparison of the top 10% to the bottom 50% is worth adding. Also, in discussing the "bottom 50%", they should begin by noting the fraction of NIH-funded PIs who have a single grant (my memory suggests ~75%) and that this combined with the modular budget means most in the bottom half are quite similar in funding levels.

3. A clearer and more extensive description of some of the methods used

a. Figure 1. The authors should provide a clearer explanation of the nature of the data in panels B and D, and how to interpret it. This is true when they use the same measures later.

b. Table 1. Make clear that for most measures the parenthetical values are percentages. Then explain what the parenthetical values represent in Age and Funding – are they ranges? Define IQR.

c. Table 2. Same Adjustments as in Table 1. Additionally, are the funding levels in some sort of inflation adjusted dollars – otherwise they seem very odd.

4. The fact that there is much greater inequality within groups than between groups is interesting and some explicit discussion of what this means would be helpful. It seems that researchers receiving more funding at later career stages may be reasonable as they on average (with speculation) may have larger networks and know the research questions that could be asked better. Thus, some inequality here may be considered acceptable. But there is not funding parity for gender or race for researchers in the workforce, which are unacceptable inequalities, although these may not be as striking as the composition of the workforce and distribution of funds to the most funded investigators. How do you interpret these results collectively – which inequalities require the highest priority to address?

5. Page 13: Regarding the graphs of income inequality overall in the US and Europe: this context was appreciated, but additional interpretation in the discussion would be useful. Funding doesn't accumulate exactly like wealth can from investments, so why do you think inequalities in income and research funding are so closely associated? Could it be a coincidence?

6. Per the authors' statement, data and code should be publicly deposited upon acceptance or earlier.

---

## [Author Response]

Essential revisions:1. Revised wording/clarifications to make sure conclusions better match the data.a. Abstract. First, stating the trend toward reversing inequality has "reversed", while technically accurate, implies return to the start of the analysis. In fact, there has been only a modest reversal, and inequality remains very strong. This sentence should be more nuanced, as should the following statement about career stages. Second the statement about women is very confusing. They state "Women continue to constitute a greater proportion of funded principal investigators, though not at parity." When I first read this, I was thinking-no chance that women are "a greater proportion", i.e., a majority of funded PIs. It would be clearer to state: "The fraction of women among funded PIs continues to increase, but they are still not at parity"

We agree with the reviewer. We have revised the abstract accordingly.

b. Figure 1 should have the Y-axis go to zero, to make it clearer just how much funding is held by the top 1%.

We respectfully disagree with the reviewer and point to the renowned graphics expert Edward Tufte. Professor Tufte wrote, “In general, in a time-series, use a baseline that shows the data not the zero point. If the zero point reasonably occurs in plotting the data, fine. But don't spend a lot of empty vertical space trying to reach down to the zero point at the cost of hiding what is going on in the data line itself. (The book, How to Lie with Statistics, is wrong on this point.) For examples, all over the place, of absent zero points in time-series, take a look at any major scientific research publication. The scientists want to show their data, not zero. The urge to contextualize the data is a good one, but context does not come from empty vertical space reaching down to zero, a number which does not even occur in a good many data sets. Instead, for context, show more data horizontally!” (See https://www.edwardtufte.com/bboard/q-and-a-fetch-msg?msg_id=00003q).

Nonetheless, we did revise Figure 1, Panel A according to the reviewer’s recommendation.

c. P. 4. The authors should note that the age range of ALL NIH funded investigators is skewing older over time-this has been analyzed by others but is worth a mention.

We agree with the reviewer and included text accordingly (lines 86-88).

d. The data in Figure 2 is very interesting, and deserves a more complete description, especially as it brings these analyses up to the current date.

We agree with the reviewer and have added text (lines 95-100).

e. P 10, Figure 3, the choice of data presentation and the way it was discussed significantly minimized the differences by career stage, race and gender. I imagine the majority of investigators in all groups are funded by a single grant at or near the modular level, and this is leveling differences – the mean/median difference supports this. I suspect that, and the authors have data to explore whether, the most highly funded individuals are very different between groups. What would the top decile look like for each group – their mean values suggest substantial differences? I'd like to see more in-depth analysis here. I think the authors have dug deeper, as they state "Women were younger, more likely to hold a PhD degree, and less likely to be principal investigators of more than the equivalent of 4 R01 grants.", but how one would assess this last factor from the data presented is not clear.

We agree with the reviewer and have added analyses of number of RPG’s per investigator (see Tables 1-5, lines 88-90 and 105-111).

f. The analysis of different institutions was stunning (Figures 6 and 7), and deserves some mention in the abstract and some more text talking about the nuances. I also wondered if the authors have analyzed by region of the country, as this is also an area that has attracted attention (e.g. analyses of Walls)?

We agree with the reviewer and added new Figures 8 (for region) and 9 (for states).

2. Additional analysesa. The analysis in Figure 1 was exceptionally interesting. A similar comparison of the top 10% to the bottom 50% is worth adding. Also, in discussing the "bottom 50%", they should begin by noting the fraction of NIH-funded PIs who have a single grant (my memory suggests ~75%) and that this combined with the modular budget means most in the bottom half are quite similar in funding levels.

We agree with the reviewer. We added data on the top decile in Figure 1, Panel A, and on the number of grants per investigator in Tables 1-5.

3. A clearer and more extensive description of some of the methods useda. Figure 1. The authors should provide a clearer explanation of the nature of the data in panels B and D, and how to interpret it. This is true when they use the same measures later.

We agree with the reviewer. We have added text (lines 58-61, 72-73).

b. Table 1. Make clear that for most measures the parenthetical values are percentages. Then explain what the parenthetical values represent in Age and Funding – are they ranges? Define IQR.

We agree with the reviewer and made these changes.

c. Table 2. Same Adjustments as in Table 1. Additionally, are the funding levels in some sort of inflation adjusted dollars – otherwise they seem very odd.

We agree with the reviewer and made these changes. We note that dollar figures are inflation-adjusted. (See also lines 187-188).

4. The fact that there is much greater inequality within groups than between groups is interesting and some explicit discussion of what this means would be helpful. It seems that researchers receiving more funding at later career stages may be reasonable as they on average (with speculation) may have larger networks and know the research questions that could be asked better. Thus, some inequality here may be considered acceptable. But there is not funding parity for gender or race for researchers in the workforce, which are unacceptable inequalities, although these may not be as striking as the composition of the workforce and distribution of funds to the most funded investigators. How do you interpret these results collectively – which inequalities require the highest priority to address?

We agree with the reviewer. Without wading into policy pronouncements (which would be inappropriate for this venue), we add some discussion (lines 171-177).

5. Page 13: Regarding the graphs of income inequality overall in the US and Europe: this context was appreciated, but additional interpretation in the discussion would be useful. Funding doesn't accumulate exactly like wealth can from investments, so why do you think inequalities in income and research funding are so closely associated? Could it be a coincidence?

We agree with the reviewer and added appropriate text (lines 142-144).

6. Per the authors' statement, data and code should be publicly deposited upon acceptance or earlier.

We agree with the reviewer. We will deposit de-identified data and code upon acceptance.